# Bacterial Culture Underestimates Lung Pathogen Detection and Infection Status in Cystic Fibrosis

Helen Gavillet,[a] Lauren Hatfield,[a] Damian Rivett,[b] Andrew Jones,[c,d] Anirban Maitra,[e] Alexander Horsley,[c,d] Christopher van der Gast[a,f]

[a]Department of Life Sciences, Manchester Metropolitan University, Manchester, United Kingdom
[b]Department of Natural Sciences, Manchester Metropolitan University, Manchester, United Kingdom
[c]Manchester Adult Cystic Fibrosis Centre, Manchester University NHS Foundation Trust, Manchester, United Kingdom
[d]Division of Infection, Immunity and Respiratory Medicine, University of Manchester, Manchester, United Kingdom
[e]Royal Manchester Children's Hospital, Manchester University NHS Foundation Trust, Manchester, United Kingdom
[f]Department of Respiratory Medicine, Northern Care Alliance NHS Foundation Trust, Salford, United Kingdom

Alexander Horsley and Christopher van der Gast are senior authors of this work.

**ABSTRACT** Microbiological surveillance of airway secretions is central to clinical care in cystic fibrosis (CF). However, the efficacy of microbiological culture, the diagnostic gold standard for pathogen detection, has been increasingly questioned. Here we compared culture with targeted quantitative PCR (QPCR) for longitudinal detection of 2 key pathogens, *Pseudomonas aeruginosa* and *Staphylococcus aureus*. Prospectively collected respiratory samples taken from 20 pediatric and 20 adult CF patients over a period of 3-years were analyzed. Patients were eligible if considered free of chronic *Pseudomonas* infection within 12-months prior to start of study. QPCR revealed high levels of infection with both pathogens not apparent from culture alone. *Pseudomonas* and *Staphylococcus* were detected by culture on at least one sampling occasion in 12 and 29 of the patients, respectively. Conversely, both pathogens were detected in all 40 patients by QPCR. Classification of infection status also significantly altered in both pediatric and adult patients, where the number of patients deemed chronically infected with *Pseudomonas* and *Staphylococcus* increased from 1 to 28 and 9 to 34, respectively. Overall, *Pseudomonas* and *Staphylococcus* infection status classification changed respectively for 36 and 27 of all patients. In no cases did molecular identification lead to a patient being in a less clinically serious infection category. Pathogen detection and infection status classification significantly increased when assessed by QPCR in comparison to culture. This could have implications for clinical care of CF patients, including accuracy of infection diagnosis, relevant and timely antibiotic selection, antimicrobial resistance development, establishment of chronic infection, and cross-infection control.

**IMPORTANCE** Chronic lung infection is the leading cause of morbidity and early mortality for people with cystic fibrosis (pwCF). Microbiological surveillance to detect lung pathogens is recommended as best practise in CF patient care. Here we studied pathogen detection in 40 pwCF over several years. We found that microbiological culture, the diagnostic gold standard, was significantly disparate to targeted culture-independent approaches for detection and determination of chronic infection status of two important pathogens in CF. Pathogen detection was significantly lower by culture and consequently infection status was also misclassified in most cases. In particular, the extent of chronic infection by both *P. aeruginosa* and *S. aureus* not realized with culture was striking. Our findings have implications for the development of infection and clinical care of pwCF. Future longitudinal studies with greater patient numbers will be needed to establish the full extent of the clinical implications indicated from this study.

**KEYWORDS** lung infection, infection status, chronic infection, QPCR, *Pseudomonas aeruginosa*, *Staphylococcus aureus*

Address correspondence to Christopher van der Gast, C.vanderGast@mmu.ac.uk.

The authors declare a conflict of interest. All authors declare support from the CF Trust. A.H. reports personal fees for advisory services (Mylan Pharmaceuticals) and educational and presentation activities (Vertex Pharmaceuticals).

Cystic fibrosis (CF) is characterized by chronic airway infection and consequent inflammation (1, 2). This starts early in life, most commonly with *Staphylococcus aureus* and *Haemophilus influenzae* infection. As time progresses, CF microbiology evolves and becomes increasingly dominated by a small number of Gram-negative pathogens rarely encountered in immunocompetent hosts. Chief among these is *Pseudomonas aeruginosa*, which increases in prevalence in adolescence and is cultured from sputum in over half of all adult CF patients (3, 4). Other important pathogens isolated in CF include *Achromobacter* spp., *Stenotrophomonas maltophilia*, and members of the *Burkholderia cepacia* complex.

Clinical diagnosis of microbial infection relies on microbiological culture as the gold standard for pathogen detection (5). Regular microbiological surveillance throughout the life of a CF patient is considered best practice (6). This is used to guide targeted treatment, provide an indication of the effectiveness of treatment against pre-existing infection, and enable discovery of recently acquired infections to allow timely eradication (6, 7). Following diagnosis of acute pulmonary exacerbation, further regular surveillance for the duration of treatment is recommended to direct antibiotic therapy (6). Furthermore, CF patients are typically categorized by infection type and attend segregated outpatient clinics to prevent cross-infection between patient cohorts colonized with, for example, *P. aeruginosa* and *B. cepacia* complex members (6, 8).

Over the last decade, culture-independent molecular microbiology techniques have become increasingly investigated. These methods are more sensitive than culture and are able to identify infections which are harder to detect using traditional microbiological culture alone (9). As such, molecular-based approaches like quantitative PCR (QPCR) have been increasingly proposed as alternatives to culture (7, 10). QPCR is well suited to this application, offering targeted organism identification and quantification with inherent high-target specificity and sensitivity, and can be performed on small amounts of material (as found in cough swabs) (7). Despite many cross-sectional studies reporting on improved sensitivity using QPCR in CF, little progress has been made in establishing such molecular techniques in clinical care, and there are few longitudinal studies which have explicitly followed individual patients over time. Furthermore, none of those studies considered the impact that more sensitive detection methods would have on infection status or clinical outcomes (see Supplementary Materials and Table S1).

In this study, respiratory samples from pediatric and adult CF patients were prospectively collected over a period of up to 3 years. These were all patients believed to be free of chronic *Pseudomonas* infection by culture prior to start of study. The aim of the study was to longitudinally compare the diagnostic gold standard of microbiological culture with targeted QPCR for detection of two prominent pathogens of concern, *P. aeruginosa* and *S. aureus*; both of which are deemed to be readily detected by culture (5). We also assessed the impact of the resulting disparity in detection methods on clinical classification and infection status (e.g., chronic or intermittent infection). Potential clinical implications for CF and for pathogen detection and surveillance more broadly are considered and discussed.

## RESULTS

Ninety pediatric and adult patients were recruited for the study. However, subsequent longitudinal analysis was restricted to patients who contributed ≥6 samples with contemporaneous diagnostic microbiology data (Fig. 1). The 20 pediatric and 20 adult patients that met those criteria contributed 327 respiratory samples over the 3-year study period, with a mean ± standard deviation (SD) of 8.2 (± 2.8) samples per patient (minimum 6 samples, maximum 20) (Table S2). Overall, 233 (71%) of samples were sputum, the remainder ($n = 94$) being cough swabs taken by clinical staff, with rates of cough swab higher in children compared to adults (42% vs 19%). Clinical characteristics of all patients are summarized in Table 1, with characteristics of individual patients presented in Table S3. Overall, this was a population with mild CF lung disease, with a mean $FEV_1$ of 80.6% predicted.

Pathogen detection by culture and QPCR approaches was compared across the patients included in the study (Fig. 2). In all instances, culture significantly underestimated pathogen detection. For *P. aeruginosa*, the mean percentage of pathogen detection (± SD) across all

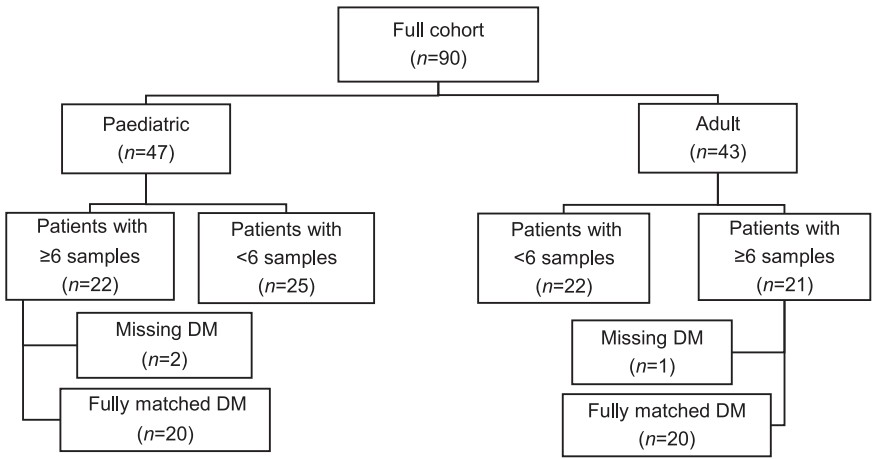

**FIG 1** Flow diagram detailing patient selection process. Only patients who contributed ≥6 samples with contemporaneous diagnostic microbiology data (DM) were included in the final analyses. Using a modification of the Leeds criteria, patients were deemed to be chronically or intermittently colonized with a given pathogen if >50% or ≤50% of samples, respectively, over the 3-year study period were positive by diagnostic microbiology or targeted QPCR. A minimum of ≥6 samples was chosen as less samples would have increased the likelihood of misclassifying infection status.

pediatric patients by culture was 8.8 ± 20.4% compared to 60.9 ± 23.4% by QPCR (Kruskal-Wallis test, $H = 24.96$, $P < 0.0001$). For *P. aeruginosa* in adults, the means were 3.3 ± 4.8% and 68.5 ± 15.7% by culture and QPCR, respectively ($H = 30.34$, $P < 0.0001$). For *S. aureus* in pediatric patients the mean detection by culture and QPCR were 21.6 ± 16.8% and 72.2 ± 20.8%, respectively ($H = 24.98$, $P < 0.0001$). Within adult patients, *S. aureus* mean detection was 44.5 ± 45.3 and 75.1 ± 25.9 by culture and QPCR, respectively ($H = 4.51$, $P = 0.034$).

More specifically, *P. aeruginosa* and *S. aureus* were detected by culture on at least one occasion in 12 (30%) and 29 (73%) of the patients, respectively. Conversely, both pathogens were detected in all 40 patients on at least one occasion by QPCR (Fig. 2). For *P. aeruginosa*, culture determined only one patient to be classified as chronically infected and a further 11 classified as intermittently infected (Fig. 2 and 3). That increased to 28 classified as chronically infected and 12 classified as intermittently infected with *P. aeruginosa* when using QPCR to identify presence of infection. For *S. aureus*, 9 patients were deemed to be chronically infected by culture compared to 34 patients classified as chronically infected by QPCR. Change in bacterial infection status classification was seen in both the adult and pediatric patients. Overall, *P. aeruginosa* and *S. aureus* infection status classification changed for 36

**TABLE 1** Summary of clinical characteristics for all patients

| Characteristics | Paediatric | Adult | All patients |
|---|---|---|---|
| No. of patients | 20 | 20 | 40 |
| Mean no. of samples per patient (±SD) | 7.0 (1.0) | 9.4 (3.5) | 8.2 (2.8) |
| No. of sample types (sputum/swab) | 81/58 | 152/36 | 233/94 |
| Sex (female/male) | 10/10 | 8/12 | 18/22 |
| Mean age (±SD)[a] | 10.5 (2.7) | 25.1 (4.6) | 18.9 (8.1) |
| Mean %FEV$_1$ (±SD) | 91.0 (15.3) | 73.0 (19.5) | 80.6 (19.9) |
| Pancreatic insufficiency (sufficient/insufficient) | 3/17 | 8/12 | 11/29 |
| CF related diabetes (Y/N) | 0/20 | 1/19 | 1/39 |
| CFTR genotype[b] | | | |
| Phe508del homozygous[c] | 9 | 7 | 16 |
| Phe508del heterozygous | 7 | 8 | 15 |
| Non-Phe508del | 4 | 5 | 9 |

[a]SD denotes standard deviation of the mean. Based on age at time of first sample for each patient.
[b]CFTR genotype - cystic fibrosis transmembrane conductance regulator genotype.
[c]Homozygous Phe508del, 2 copies of the Phe508del gene mutation. Heterozygous Phe508del, single copy of this mutation plus another mutation.

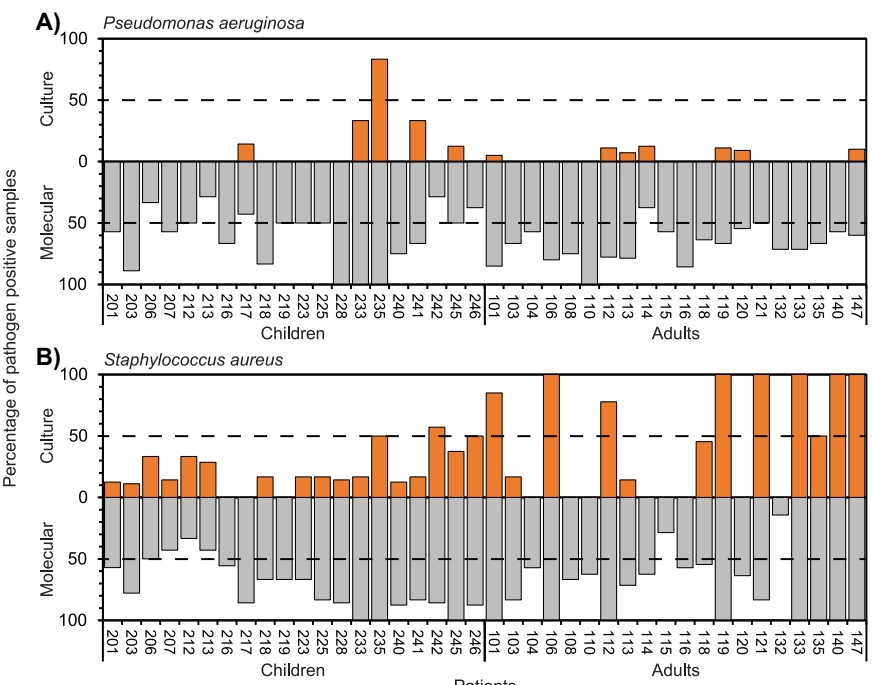

**FIG 2** Pathogen detection by conventional culture and molecular-based approaches in pediatric and adult cystic fibrosis patients. Given for each patient, is the percentage of respiratory samples over the 3-year study duration that were culture (orange) or QPCR (gray) positive for (A) *Pseudomonas aeruginosa* and (B) *Staphylococcus aureus*. In each instance, the dashed line denotes the threshold for chronic (>50%) or intermittent (≤50%) colonization. Numbers on the *x* axis represent individual patient study numbers.

and 27 out of all patients, respectively. In no cases did molecular identification lead to a patient being allocated to a negative or intermittent infection category when culture placed them with an intermittent or chronically infected classification (Fig. 3).

There was only a single instance of a greater percentage of pathogen positive samples detected by culture than by QPCR; specifically, *S. aureus* in patient 121 (Fig. 2B). That patient was however still deemed to be chronically colonized with *S. aureus* irrespective of method of detection. Out of the 18 samples culture positive for *P. aeruginosa* samples, only 5 of those did not have corresponding detection by QPCR. For *S. aureus*, from the 121 culture positive samples, 8 did not have matching molecular detection.

Pathogen abundance was assessed to ascertain whether it contributed to the underestimation of pathogen detection by culture; whereby low bacterial cell abundances could contribute to false negatives by culture. However, abundance derived by QPCR for both pathogens was relatively high across all samples (Fig. S1). The mean abundance ($\pm$ SD) for *P. aeruginosa* in pediatric and adult patients was $1.91 \times 10^9 \pm 1.11 \times 10^1$ CFU (CFU) mL$^{-1}$ equivalents and $4.75 \times 10^8 \pm 1.34 \times 10^1$ CFU mL$^{-1}$ equivalents, respectively. For *S. aureus*, mean abundances were $7.37 \times 10^8 \pm 2.54 \times 10^1$ and $1.85 \times 10^8 \pm 1.14 \times 10^1$ CFU mL$^{-1}$ equivalents for pediatric and adult patients, respectively. Similarly, the influence of respiratory sample type was assessed, with an expectation that cough swabs would have lower pathogen abundances when compared to sputum as a source sample. With the expectation of *P. aeruginosa* in pediatric samples, there was no significant differences in pathogen abundances when derived from sputum or cough swab samples (Fig. S2).

## DISCUSSION

In this study, we compared the detection of key pathogens on repeated samples from the same cohort of CF children and adults over several years. Since the patients were considered free of chronic *Pseudomonas* infection by culture, we have focused on the identification of this pathogen along with *S. aureus* in sputum and cough swabs collected at the same time as diagnostic microbiology. Both pathogens are broadly accepted as readily detectable

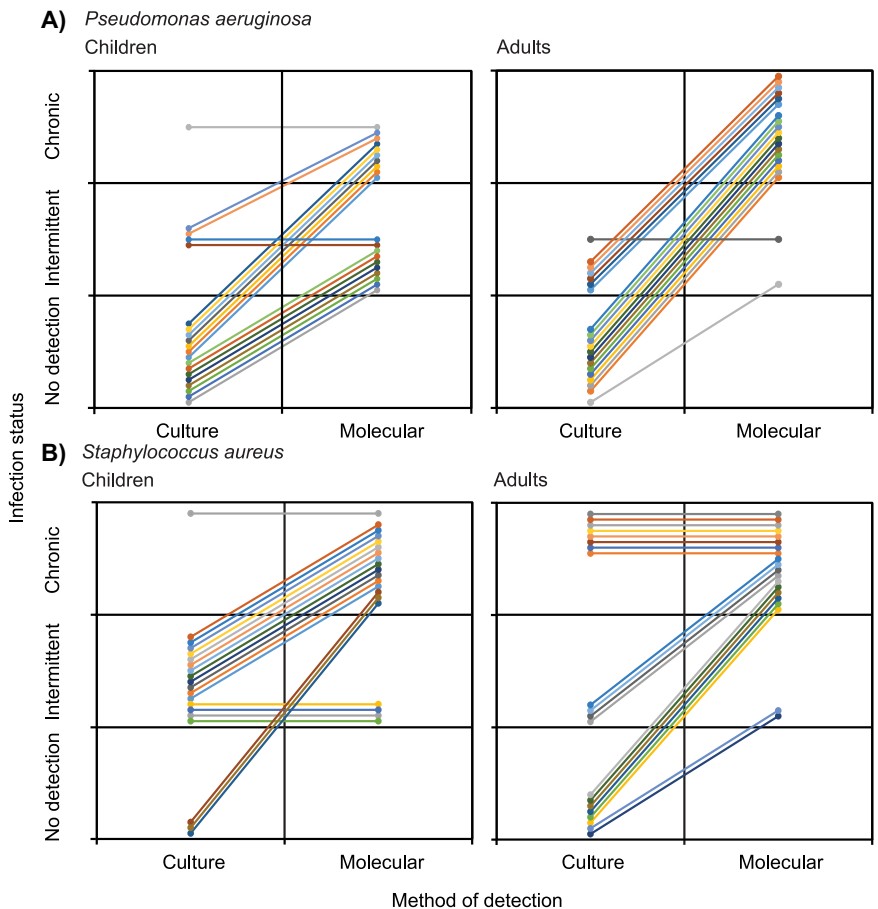

**FIG 3** Changes in pathogen infection status classification from when defined by diagnostic culture to then by molecular-based detection in pediatric and adult patients. Given are changes in infection status classification for (A) *Pseudomonas aeruginosa* and (B) *Staphylococcus aureus* in each of the children and adult CF patients. In each instance, colored lines represent individual patients.

by culture-based diagnostic microbiology (5, 11). Strikingly, detection of both *P. aeruginosa* and *S. aureus* was significantly lower by culture, which in, many instances, did not detect either pathogen despite these being found repeatedly by targeted QPCR (Fig. 2). Consequently, the infection status of patients was also altered in the majority of instances (Fig. 3), with potential implications for understanding of infection in CF and clinical management of emergent infections.

Current infection surveillance approaches in CF are guided by classical aerobic culture-based diagnostic microbiology (5, 6), and these data are collated across CF centers and used to inform national annual CF patient registry reports (3, 4). Based on these culture-based microbiology results, patients are also cohorted into groups and segregated to avoid cross-infection (6, 8). In this study, patients were specifically drawn from *Pseudomonas*-free clinics, containing patients with no new growth of *P. aeruginosa* for at least 12 months and considered free of chronic infection by this pathogen. Although it was accepted that some would subsequently become infected by this pathogen over the 3-year study period, culture indicated all patients were free of chronic infection throughout the study, with only one exception (Fig. 2). Conversely, QPCR indicated that 28 (70%) of all patients were chronically infected with *P. aeruginosa*, and the remainder intermittently infected. Based on culture-based reporting alone, none of the pediatric and adult patients would have been redirected into *P. aeruginosa* based outpatient clinics.

In the case of *Pseudomonas* in particular, this has implications for the treatments that patients are offered. Eradication treatments, targeted oral or intravenous therapies for exacerbations, and long-term inhaled anti-pseudomonal antibiotics are not prescribed unless *P.*

*aeruginosa* is isolated on culture. Our data would indicate that this may significantly underestimate the true prevalence of chronic infection. Suboptimal antimicrobial therapy could further contribute to continued infection, and an increase likelihood of developing antimicrobial resistance or transitioning to chronic infection with time (12). Importantly, underestimation or non-detection may miss the critical period in early *P. aeruginosa* infection typically used as an ideal time frame for targeted eradication therapy to prevent establishment of chronic infection (13).

Over the last 10 years or so, molecular microbiology techniques, have challenged our fundamental understanding of CF airway pathogen epidemiology (14); with previous literature focused on QPCR in CF summarized in the supplemental materials and Table S1. In support of our findings, a CF lung microbiome study of 297 patients attending 13 CF centers in Europe and the USA, representing a broad cross-section of respiratory disease, found *P. aeruginosa* in all patients and *S. aureus* in the majority of those patients (93%) (15). However, there have been surprising few studies though that have addressed this issue in longitudinal sampling. Most notably, in a study of 80 Danish CF patients, whole genome sequencing approaches and positive culture history of *P. aeruginosa* infections demonstrated that the same clone type could persist from cultures taken before and after a judged *Pseudomonas*-free eradication period in many of those patients (16). This suggested that the original *Pseudomonas* strain had not been cleared, and was missed by routine culture-based surveillance, as we have shown. Similar to the current study, Héry-Arnaud et al. (17) looked at patients apparently free of *Pseudomonas* and compared QPCR with culture. They showed that QPCR would often become positive before culture and showed high rates of molecular identification of *Pseudomonas* in those in whom it was never cultured (17). Results were presented in aggregate however, and the impact on classification of microbiological status of individual patients was not reported.

Although methodological strengths and weaknesses of molecular and culture detection methods have been reviewed by others (7, 10), it is important to consider potential factors that might contribute to the observed disparity between the sets of outcomes. For instance, low pathogen abundance within a sample or low abundance resulting from different respiratory sample types (e.g., cough swab versus sputum) could be factors leading to misdetection by culture (7, 10). However, given the relatively high abundances observed for both pathogens by QPCR across all patients this is unlikely to have been a major factor here (Fig. S1 and S2). It is also possible that artifacts of cell death were amplified by QPCR, creating false positives. Again, this seems unlikely to be a major or constant factor given the observed abundances across all patients over the 3-year study period. Viable but not culturable (VBNC) status may explain to some degree why these readily culturable pathogens were undetected in many cases (17–19). VBNC is believed to be a protective state in response to some form of environmental stressor, including changes from aerobic or anaerobic conditions, prevalence of energy and nutrient sources, or as a defense mechanism against antibiotics. Pertinent to this study, the ability of antibiotics to induce a VBNC state have been proposed from studies of *S. aureus* and *P. aeruginosa*, which may also support pathogen survival to extended and repeated antibiotic interventions (18, 19). The clinical impact of bacteria in such states on either current or future health remains to be established. In our study, there were a few instances where there was detection by culture but not by QPCR. However, it is possible for culture to misidentify pathogens and generate false positives. For example, other *Staphylococcus* species can be mistaken for *S. aureus* in routine testing (11, 20).

Microbiological surveillance to detect pathogens is the foundation of CF clinical care and treatment. Our longitudinal data substantially builds on previous observations to demonstrate that the diagnostic gold standard of microbiological detection by culture may significantly underestimate prevalence (Table S1). Crucially, we have further shown how classification of a patient's infection status would change using findings from QPCR in comparison with culture even for what are believed to be readily culturable pathogens from pediatric and adult patients. Despite following patients over several years, the current study was limited to 40 patients. To further advance this work, future studies will need to have larger patient numbers representative of the wider CF population, encompassing, for example, differing disease

severities, disease states, and exacerbation and infection histories. That said, it remains that our findings have potential implications for understanding the evolution of infection and the clinical care of patients. This issue will become even more important in CF patients treated with highly effective cystic fibrosis transmembrane (CFTR) modulators. These treatments often lead to dramatic reductions in sputum volumes, increasing our reliance on culture of other specimens than sputum, such as cough swabs, with known reduced sensitivity in detection of pathogens. Thus, molecular detection may have an increasing potential role for detection of potential CF pathogens, but there is a need to establish its relevance in comparison with current culture-based techniques (21). These findings also have implications beyond CF, since microbiological detection and surveillance are critical to guide therapeutics in other airways diseases. As targeted molecular identification becomes quicker and cheaper, it is likely to become increasingly deployed in clinical scenarios. Subsequently, treatment approaches developed in response to conventional clinical microbiology need reevaluation to deal with these more sensitive techniques. Again, future longitudinal studies with larger patient numbers will be needed to establish the extent of the clinical implications arising from this study. Moreover, such studies will be required to thoroughly assess targeted QPCR approaches before they could be approved for routine and widespread diagnostic use in clinical settings.

## MATERIALS AND METHODS

**Study and patient sampling.** Patients were recruited as part of a longitudinal observational study of adults and children with CF cared for at two different CF centers (22). Adults were recruited from the Manchester Adult CF Centre (center patient population $n = 466$ patients), and children from the Royal Manchester Children's Hospital ($n = 342$ patients). Patients were required to be at least 5 years old, with a $FEV_1$ of >50% predicted at study entry. Patients were drawn from *Pseudomonas*-free clinics and were eligible if considered free of chronic infection with *P. aeruginosa* and with no new growth of this pathogen for at least 12 months using diagnostic culture approaches and as defined by the Leeds criteria (23). Patients or parents/guardians provided written informed consent and children provided assent. This study was reviewed and approved by the NHS Research Ethic Committee North West, Lancaster (Ref 14/NW/1195).

Patients were assessed at their usual clinic appointments by their regular clinical team and included routine and emergency visits. Spirometry was performed by the usual clinical team. Normal ranges for spirometry were those from the Global Lung Initiative (24). At study entry, 85% of adults (17/20) were on long-term azithromycin treatment, while one patient (patient 110) was receiving long-term colistin and the two remaining patients (133 and 135) were not prescribed any long-term antibiotics. Seventy percent of pediatric patients (14/20) were also on long-term azithromycin, the remaining 6 (201, 206, 212, 216, 228 and 241) were not prescribed any long-term antibiotics (Table S2 and S3). Sputum or cough swab samples were taken at each clinic visit for diagnostic and molecular microbiology. All sputum samples were spontaneously expectorated. Molecular microbiology samples were transported to the lab within 3 h and stored at −80℃ prior to DNA extraction and PCR (25, 26). Sputum samples were mixed and weighed prior to splitting and storage. To assess longitudinal microbiology, only patients with a minimum of six respiratory samples along with accompanying full diagnostic microbiology data were included (Fig. 1 and Table 1).

**Diagnostic microbiology.** Diagnostic culture-based microbiology data was provided by the microbiology service within Manchester University NHS Foundation Trust who perform microbiology testing for the participating centers, in line with international guidance and standards (5, 6).

**DNA Extraction and QPCR.** Nucleic acid extraction was performed on sputum and cough swab samples as previously described, with a modification for the latter sample type (27). As an alternative to the wash stage for sputum, cough swabs were saturated in sterile phosphate buffer solution for 5 min, then squeezed with sterile tweezers to extract as much material as possible. The resulting solution was then introduced at the bead beating stage and the protocol continued as normal as for the sputum samples thereafter.

It was not within the scope of the current study to design and assess new primers for targeted QPCR against *P. aeruginosa* and *S. aureus*, as many effective primer sets have been developed and evaluated over at least the last 2 decades; for example, see Table S1 for *P. aeruginosa* based primers. Instead, the approach here was to use primer sets already established as highly effective.

All QPCR was performed on a Bio-Rad CFX connect machine (Bio-Rad, Deeside, UK), Each plate had a blank consisting of the master mix, probes, primers and water, and pure strains of each pathogen were run on each plate in a 10-fold dilution as both a positive control and a detection standard.

**P. aeruginosa – oprL.** While there are several targets for *P. aeruginosa* amplification, studies vary in their usage. Since development in the late 1990s, the *oprL* gene target has consistently shown good sensitivity and specificity across multiple studies (28). Multiple studies have utilized the *oprL* gene to identify *P. aeruginosa* in CF sputum and swabs (See Table S1). While some studies have further characterized the *P. aeruginosa* positive samples using confirmatory QPCR for the *gyrB/ecfX* gene targets (See Table S1), the sensitivity and specificity of *oprL* alone (100% and 70–75% respectively) (28, 29) was sufficient for this study, particularly in light of the further expense needed for confirmation with other gene targets. For this study, the average $r^2$ (efficiency) value was 0.97 across 5 runs (min: 0.96, max: 0.99). The $r^2$ in this study sat between 0.99 and 0.96 for all five runs.

*P. aeruginosa* detection targeting the *oprL* was performed using the following forward primer

(0.2 $\mu$L from 100 $\mu$M stock) CGAGTACAACATGGCTCTGG, reverse primer (0.2 $\mu$L from 100 $\mu$M stock) ACCGGACGCTCTTTACCATA, and probe (0.2 $\mu$L from 100 $\mu$M stock) FAM–CCTGCAGCACC-AGGTAGCGC-TAMRA (30, 31). A further 2 $\mu$L of DNA, 20 $\mu$L of TaqMan Gene expression Mastermix (Applied Biosystems) and 17.4 $\mu$L of molecular grade water was added to a final volume of 40 $\mu$L. Samples underwent an initial denaturation for 10 min at 950C, followed by 40 cycles consisiting of 10 s denaturation at 950C, 30 s of annealing at 580C, and a 1 min elongation at 720C.

**S. aureus – nuc.** The *nuc* gene used for *S. aureus* detection was developed 1992 (32). The gene is highly specific for *S. aureus* and is not found in other key staphylococcal strains (33). The *nuc* gene is a well-established *S. aureus* QPCR primer, with a sensitivity and specificity of around 98% and 100% respectively (34–37). For this study, the average $r^2$ (efficiency) value for this target was 0.97 over 4 runs (min: 0.94, max: 1).

*S. aureus* detection targeting the *nuc* was performed using the following forward primer (0.06 $\mu$L from a 100 $\mu$M stock) CGCTACTAGTTGCTTAGTGTTAACTTTAGTTG, reverse primer (0.06 $\mu$L from a 100 $\mu$L stock) TGCACTATATACTGTTGGATCTTCAGAA, and probe (0.02 $\mu$L from 100 $\mu$M stock) FAM-TGCATCACAAACAGA TAACGGCGTAAATAGAAG-TAMRA (34). A further 2 $\mu$L of DNA, 10 $\mu$L of TaqMan Gene expression Mastermix and 7.86 $\mu$L of molecular grade water was added to a final volume of 20 $\mu$L. Samples underwent an initial denaturation for 10 min at 950C, followed by 40 cycles consisiting of 30 s denaturation at 950C, 30 s of annealing at 600C, and a 1 min elongation at 72°C.

**Data analysis.** For longitudinal analysis, this was restricted to patients who contributed ≥6 samples with contemporaneous diagnostic microbiology data (Fig. 1). Using a modification of the Leeds criteria (23), patients were deemed to be chronically or intermittently colonized with a given pathogen if >50% or ≤50% of samples, respectively, over the 3-year study period were positive by diagnostic microbiology or targeted QPCR. A minimum of 6 samples were chosen as fewer samples would have increased the likelihood of misclassifying infection status. Summary statistics, including means and standard deviations (SD), were calculated using XLSTAT v2018.1 (Addinsoft). Kruskal-Wallis analyses in conjunction with *post hoc* Dunn tests were performed in XLSTAT. Significance was set at $P < 0.05$.

**Data availability.** Culture and QPCR data from the study has been deposited at figshare.com under https://doi.org/10.6084/m9.figshare.14483394.v2.

## SUPPLEMENTAL MATERIAL

Supplemental material is available online only.
**SUPPLEMENTAL FILE 1**, PDF file, 1.3 MB.

## ACKNOWLEDGMENTS

We thank the patients who took part in this and the clinical teams who supported this work. This work was supported by the CF Trust (VIA 045), the NIHR (CS012-13), and the North West Lung Centre. This work was also supported by the NIHR Manchester Clinical Research Facility.

The funders had no role in study design, data collection, interpretation, or the decision to submit the work for publication. All authors declare support from the CF Trust. A.H. reports personal fees for advisory services (Mylan Pharmaceuticals) and educational and presentation activities (Vertex Pharmaceuticals).

C.v.d.G., A.H., and H.G. conceived the study. H.G., L.H., and D.R. performed sample processing and analysis. H.G., D.R., and C.v.d.G. performed data and statistical analysis. A.H., A.M., and A.J. were responsible for sample collection, clinical care records and documentation. H.G., A.H., and C.v.d.G. verified the underlying data. H.G., A.H., and C.v.d.G. were responsible for the creation of the original draft of the manuscript. All authors contributed to the development of the final manuscript. C.v.d.G. and A.H. are guarantors of this work. All authors read and approved the final manuscript.

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
