## [Reviewer comments · Microbiology Spectrum]

Microbiology Spectrum

Bacterial culture underestimates lung pathogen detection and infection status in cystic fibrosis

Helen Gavillet, Lauren Hatfield, Damian Rivett, Andrew Jones, Anirban Maitra, Alexander Horsley, and Christopher van der Gast

Corresponding Author(s): Christopher van der Gast, Manchester Metropolitan University

Review Timeline:

Submission Date:	February 2, 2022
Editorial Decision:	March 8, 2022
Revision Received:	May 18, 2022
Editorial Decision:	June 22, 2022
Revision Received:	July 9, 2022
Accepted:	July 31, 2022

Editor: Joanna Goldberg

Reviewer(s): The reviewers have opted to remain anonymous.

Transaction Report:

DOI: <https://doi.org/10.1128/spectrum.00419-22>

March 8, 2022

Prof. Christopher van der Gast
Manchester Metropolitan University
Department of Life Sciences
John Dalton Building
Chester Street
Manchester M1 5GD
United Kingdom

Re: Spectrum00419-22 (Greater than anticipated levels of chronic lung infection found in paediatric and adult cystic fibrosis patients)

Dear Prof. Christopher van der Gast:

Thank you for submitting your manuscript to Microbiology Spectrum. You will see that both the reviewers found this paper of interest. However both also had a number of important comments on your manuscript. I also read your paper in detail and agree with their critiques. I also would like you to (1) add in additional information as how sputum was collected (whether it was induced or expectorated). You mention this briefly, but I believe this could be made clearer. I also think you should (2) describe the primers and the conditions and the controls for the QPCR in the Materials and Methods in the main text. I believe this would be of interest to the community and it gets lost in the supplementary material.

Link Not Available

Sincerely,

Joanna Goldberg

Journals Department
Reviewer comments:

Reviewer #1 (Comments for the Author):

Review of Gavillet and Colleagues' manuscript "Greater than anticipated levels of chronic lung infection found in paediatric and adult cystic fibrosis patients" which was submitted for consideration to Microbiology Spectrum.

Summary:

In this study, the authors examined a cohort of 40 children and adults with cystic fibrosis in Manchester, England. Patients were eligible for the study if they were free of chronic *Pseudomonas aeruginosa* infection. Chronic *P. aeruginosa* was defined as positivity for *P. aeruginosa* in > 50% of cultures in the preceding 12 months, although a statement in the discussion mentioned being free of any new *P. aeruginosa*. The authors additionally required subjects be at least 5 years old and have at least 6 follow up cultures. The rationale for excluding subjects under age 5 was not discussed. The follow up time ranged up to four years. The patients were enrolled in this study between 2014 and 2018, a time in which some CFTR modulator drugs may have been available, but not elxacaftor/tezacaftor/ivacaftor. The patients generally had good lung function.

Within this cohort, the authors compared two methods for detecting the canonical CF pathogens *P. aeruginosa* and *Staphylococcus aureus*: standard culture in the clinical microbiology laboratory versus Quantitative PCR, which was performed in their research laboratory. The Q-PCR assay used the previously validated PCR targets *nuc* (*S. aureus*) and *oprL* (*P. aeruginosa*). They found that the Q-PCR method resulted more cultures being categorically positive for each of the two pathogens. If the Q-PCR method was used to declare positivity for each pathogen, 38 of the 40 patients would be classified as having chronic infection with each of the pathogens.

The authors compared their findings to previously published studies with similar study design and note that that their study tracked individual patients rather than group averages, thus showing "impact on clinical status for individual patients."

Critique:

The findings in the study are credible and add to a growing body of literature showing that molecular diagnostics have superior sensitivity for cystic fibrosis pathogens when compared to traditional culture methods. The data supplement provided a literature review, culture timelines for each patient, the number of positive tests per subject, and the abundance of each pathogen. I think this provided both context and data transparency.

My concerns are mostly in how the data are discussed. The effect size in this study could be higher than predicted because of selecting patients who are culture negative at baseline and by testing single subjects repeatedly. I think that the study would benefit from some changes to the discussion that acknowledge the potential for selection bias as well as some of the inherent limitations of molecular diagnostics.

Specific comments

1. Title - The word "levels" in the title was not specific. I thought it meant greater abundance or titer of the pathogens in these patients.

While quantitative PCR was used as the method in this study, the primary study outcome was how patients were classified for chronicity of infection rather than the abundance of pathogen in samples. Because the study title does not refer to the comparison of diagnostic methods or the primary outcome, I would suggest a different title. Possibly: Q-PCR detection of cystic fibrosis pathogens increases the diagnosis of chronic lung infections.

2. The phrase "status disparity" in the running title sounds like a social comparison rather than a comparison of methods for classifying infections.

3. Subject Selection. There was some ambiguity in how the baseline infection status was defined, especially in comparing Discussion Line 157 with methods line 240.

Line 157 says patients were drawn from *Pseudomonas*-free clinics, containing patients with no new growth of *P. aeruginosa* for at least 12 months. Line 240 says that patients were eligible if considered free of "chronic infection" as defined in reference 23. Could the authors provide baseline *S. aureus* and *P. aeruginosa* culture data for the 12 months preceding the study?

4. Subject Selection. Figure 1 did not give the size of the center. It would be better to know how many subjects were excluded due to chronic *P. aeruginosa* or other exclusion criteria mentioned in the methods section such as age. The authors should also explain the rationale for excluding children under age 5.

5. Subject Characteristics. Table 1 did not report CFTR modulator therapy, which might affect quantitative recovery of pathogens or maybe even the incident detection of the pathogens. Azithromycin and Colistin were mentioned in the text. It was not given whether patients were receiving oral anti-*Staphylococcal* medications or other anti-*Pseudomonas* medications like inhaled tobramycin or aztreonam. Baseline infections for the two pathogens were not provided.

6. Outcome definition. Because the main outcome of their assay is categorical - whether *S. aureus* or *P. aeruginosa* were detected, the authors need to provide information about how they defined a positive test. Was there a Ct value they considered

positive? What was the limit of detection?

7. Quantitative results for the Q-PCR assay were presented in the supplement, given in CFU per mL equivalents. The methods section indicates the Q-PCR assay used pure strains as standards. I assume this is how CFU equivalents was determined. It is not clear how the sample volume was determined for cough swabs. Could the authors describe the calculation in greater detail?

8. Were there quantitative differences in the Q-PCR assay when comparing culture-positive and culture-negative samples?

9. The boxplots shown in the supplement were confusing. Perhaps the authors were trying to fit logarithmically-distributed data with whiskers that were defined from a linear-range calculation. The number of points above the upper whiskers in Figure S2 were far more than the number of points below the lower whiskers. In several plots in S1 and in the upper left-hand plot of S2 (*P. aeruginosa* in Pediatric Sputum), the upper whisker was below the 75th Percentile. Individual data points on this figure were on top of one another so that they couldn't be counted, especially in adults with sputum. Given the small number of patients, these points could be distributed better horizontally. I would suggest using GraphPad Prism or the beeswarm package in R so that individual data points are more apparent.

10. S1 figure was helpful because it gave the number of positive tests per subject. The total number of tests per subject can be found in table S2. The number of data points divided by the number of tests was equal to the height of the gray bars in Figure 2. The boxplots did not add to the figures for individual patients, especially when there were only 2 or 3 positive tests. It would be better if data points were outlined to make them easier to count. The limit of detection for the assay should be stated in the legend or on the figure. The S2 legend should explicitly state that it shows quantitation for positive tests only.

11. S2 figure - Pediatric *P. aeruginosa* panel. The legend gave mean {plus minus} an error estimate (I assume SD). The SD was larger than the mean, consistent with a skewed distribution. I don't think a statistical comparison is essential here, but the data distributions appeared different to me. The 75th percentile for cough swab was roughly the same as the 25th percentile for sputum. Since these data were not normally distributed, I suggest using Wilcoxon rank-sum test to compare these distributions rather than the Welch's t-test, if the authors want to make a statistical comparison.

12. Figure 2 displayed the percentage of positive samples as a stacked bar graph, stratified by patient, with children and adults divided on the x-axis. It was a nice visualization. I would recommend removing the minus signs on the y-axis for the molecular diagnostics.

13. Figure 3 provides the same data as Figure 2 after applying > 0% and > 50% thresholds. It is not clear whether this classification scheme was based on test positivity over the entire duration of the study, or the final year of the study.

14. The Figure 3 legend says "Changes in pathogen infection status from diagnostic culture to molecular-based detection." I think this could be rephrased, because the test didn't change whether the patient is actually infected - it changes how the authors would classify the infection.

15. Results Lines 108. The authors make head-to-head comparisons of Q-PCR versus culture in terms of test positivity for the entire set of observations. The results are clearly different, but I would suggest using McNemar's test rather than Welch's t-test because the outcome they present is dichotomous (positive or negative) and the samples were paired.

16. Discussion Line 206 - There were a few instances where there was detection by culture but not by Q-PCR. Could the culture isolates be amplified by Q-PCR? Were there mutations affecting the ability to amplify the target genes?

17. The work is interesting and important, but there were some limitations that should be mentioned in the discussion:

a. While PCR testing allows detection of these pathogens, it has some limitations, like not allowing for antibiotic susceptibility testing.

b. PCR testing without sequencing does not allow the infecting strain to be identified and tracked. Culture methods would facilitate strain tracking.

c. The study did not enroll control subjects, making it hard to know how abnormal the CF measurements are versus the healthy population. It is possible that control individuals, under environmental exposure to *P. aeruginosa* would test positive by Q-PCR, yet they do not develop symptoms because their airways or immune systems effectively kill incident *P. aeruginosa*.

d. There is some potential for selection bias here, as the subjects studied by the authors were known not to be chronically infected. Thus, the patient selection enhances the apparent effect of Q-PCR detection. It would be good to know how many subjects in the center were not eligible for inclusion because they were already classified as chronically infected.

e. The supplement makes the claim that this study has "impact on clinical status for individual patients." I think that is an overstatement, as the authors provide no evidence that the clinical status (lung function, symptoms, exacerbations, etc) for individual patients has changed by virtue of this method of testing. It would be fair to say that changing the method of testing would impact how chronic infection status is classified.

Reviewer #2 (Comments for the Author):

In the study, the investigators collected sputum or oropharyngeal swabs from children (n=20) and adults (n=20) with cystic fibrosis with at least 6 samples collected over 3 years and compare detection of *Pseudomonas aeruginosa* and *Staphylococcus aureus* by culture versus qPCR. They found that qPCR detected these bacterial significantly more frequently, resulting in reclassification of infection status (negative, intermittent, chronically infected) in most patients. They conclude that this may have implications for clinical care, antibiotic use and infection control for people with CF.

Major comments

1. Clinical implications are unclear: The results of this study suggest that a large majority of people with CF are misclassified with underappreciation of chronic infection status. In children, based on culture, only one participant was classified as chronically infected with Pa whereas 10 (50%) were classified as chronic based on PCR (in adults, 0% by culture vs. 90% by PCR). While the authors note that qPCR has been found to be more sensitive than culture in prior studies (as they found), they note that clinical outcomes of these findings have not been studied. However, this current study doesn't add to the understanding of the clinical impact - was there a difference in exacerbation rate, lung function change for example in those who are identified as chronically infected with Pa by qPCR? For those who were intermittently infected with Pa - and presumably treated with inhaled antibiotics- was there a difference noted in the following culture (decrease in qPCR)?
2. Given the requirement that patients were negative for Pa at study entry, the patient group represents relatively healthy CF cohort, particularly for adults. Consistent with this, almost 50% of adults were pancreatic sufficient. What criteria were used for CF diagnosis? Did all patients have sweat chloride > 60? Were those with CFTR-related disorder included? Additional genotype information such as # with residual function mutations would be helpful in understanding the patient cohort. Were any on CFTR modulators during the study?
3. While information about exacerbation status is provided in the supplement, was this considered in the analysis? What about acute antibiotic use?
4. Figure S2 does not seem to match the results states. In the pediatric Pa panel, the abundance detected in OP appears lower than in sputum - the mean abundance in the legend and text says "7.93x10¹⁰ {plus minus} 1.86x10¹¹ (n = 24) and 5.38x10¹⁰ {plus minus} 2.14x10¹¹" for Pa in sputum versus OP. However visually, the mean of OP looks closer to 10⁷ vs 10⁹ for sputum. Similarly, the other panels don't seem to match the numbers given although OP / sputum look more similar in these groups. It's also interesting that the pediatric Pa quantities look higher than the adult Pa cultures - possibly due to more mild disease aggressiveness in adults.
5. Figure S1 is a helpful graph. For Pa (less so for Sa), it appears that many patients/ samples had lower levels of Pa. For those with low Pa detected (e.g. < 10⁸) was qPCR detection more likely to be concordant with culture?

Minor comments:

1. In the supplement, table S2, dates are provided for samples collected from individual patients. Suggest changing this to baseline, follow-up time in months rather than exact dates to avoid the risk of identifying individuals.
2. Table S1 is an outstanding summary of the current literature and the attending to detail is appreciated.
3. The use of %'s in addition to numbers throughout the results section would help the reader.
4. Figure S2: Why are the individual dots separated along the x-axis within sample groups?

Staff Comments:

Preparing Revision Guidelines

Please return the manuscript within 60 days; if you cannot complete the modification within this time period, please contact me. If you do not wish to modify the manuscript and prefer to submit it to another journal, please notify me of your decision immediately so that the manuscript may be formally withdrawn from consideration by Microbiology Spectrum.

Response to Reviewers

Spectrum00419-22 (Greater than anticipated levels of chronic lung infection found in paediatric and adult cystic fibrosis patients)

Dear Professor Goldberg,

We thank you and the reviewers for the time taken to review our manuscript. We have addressed the comments and suggestions from yourself and the reviewers in turn below. For clarity, reviewer comments are in italics with our response directly below a given comment. Starting with “**Response:** ...” in each instance. Changes made in the Marked-Up Manuscript are in red coloured text. Page and Line numbers used below refer to those in the Marked-Up Manuscript.

Editor comments:

I also would like you to (1) add in additional information as how sputum was collected (whether it was induced or expectorated). You mention this briefly, but I believe this could be made cleared. I also think you should (2) describe the primers and the conditions and the controls for the QPCR in the Materials and Methods in the main text. I believe this would be of interest to the community and it gets lost in the supplementary material.

Response: As directed, we have (1) added that all sputum samples were expectorated at Page 10 Line 254. (2) Moved QPCR text from Supplementary Material and inserted into the Materials and Methods starting at P10, L272.

Reviewer #1 comments:

The findings in the study are credible and add to a growing body of literature showing that molecular diagnostics have superior sensitivity for cystic fibrosis pathogens when compared to traditional culture methods. The data supplement provided a literature review, culture timelines for each patient, the number of positive tests per subject, and the abundance of each pathogen. I think this provided both context and data transparency.

Response: We thank the reviewer for recognising positives of this study and the importance of the literature review and our approach to data transparency. On the literature review, we specifically included this to highlight: (1) what has been done previously, and (2) how our study was novel and differed from those previous studies. This was to help potential reviewers understand how this work sat within the existing literature and appreciate how it was novel in that respect. We also appreciate the reviewer recognising our commitment to data transparency. We have also made the underpinning data available by depositing at Figshare.com (Data availability P12, L324).

My concerns are mostly in how the data are discussed. The effect size in this study could be higher than predicted because of selecting patients who are culture negative at baseline and by testing single subjects repeatedly. I think that the study would benefit from some changes to the discussion that acknowledge the potential for selection bias as well as some of the inherent limitations of molecular diagnostics.

Response: There are a number of points here which are addressed in turn. Regarding “*The effect size in this study could be higher than predicted because of selecting patients who are culture negative at baseline and by testing single subjects repeatedly*”. We are not quite sure what points the reviewer is making here. To be clear, it was the observation that these mild disease paediatric and adult patients, deemed not to be chronically colonised by *P. aeruginosa* by diagnostic culture, were found to be mostly chronically colonised or at least intermittently colonised when targeted QPCR was used is what makes our findings surprising and remarkable. We did not and could not know this *a priori*. Of course, we do not know to what the full extent of clinical implications of this finding will be. In the discussion we do discuss and consider possible infection-based clinical implications. But are careful not to overclaim and conclude the manuscript by

stating clearly: P9, L229: “Again, future longitudinal studies with larger patient numbers will be needed to establish the extent of the clinical implications arising from this study. Moreover, such studies will be required to thoroughly assess targeted QPCR approaches before they could be approved for routine and widespread diagnostic use in clinical settings”. We have tempered language further by amending the following: Introduction P4, L81: we have replaced investigated with ‘considered’. So sentence now reads: “Furthermore, none of those studies considered the impact that more sensitive detection methods would have on infection status or clinical outcomes”. At P4, L90 at the end of the introduction the sentence now reads “Potential clinical implications for CF and for pathogen detection and surveillance more broadly are considered and discussed”. Sentence at P7, L151 amended to include word ‘potential. Now reads: “...with potential implications for understanding of infection in CF and clinical management of emergent infections”. Sentence in Discussion at P9, L218 amended to include ‘potential’. Now reads: “That said, it remains that our findings have potential implications for understanding the evolution of infection and the clinical care of patients”.

With regard to “*testing single subjects repeatedly*”, this was a longitudinal study with a requirement to assess infection status (e.g. chronic, intermittent, or free). We are not sure how one would do that without sampling subjects repeatedly over time? With regards to “*I think that the study would benefit from some changes to the discussion that acknowledge the potential for selection bias as well as some of the inherent limitations of molecular diagnostics*”. We don’t believe there is selection bias as the reviewer indicates. That said, and to help the reader to better understand the cohorts, we have created a new Table in the Supplementary Materials (Table S3) that details the clinical characteristics of individual patients, including individual CFTR genotypes, lung function, etc. As noted above, data transparency is important to us. On potential methodological factors, we do acknowledge and discuss this in the Discussion from P8, L191 in the passage of text that starts “Although methodological strengths and weaknesses of molecular and culture detection methods have been reviewed by others (7, 10) it is important to consider potential factors that might contribute to the observed disparity between”. As pros and cons of methods have been extensively reviewed and discussed in detail elsewhere (including in reviews papers – references 7 and 10) we felt this was sufficient and kept to points we felt pertinent to this study.

Specific comments

1. Title - The word “levels” in the title was not specific. I thought it meant greater abundance or titer of the pathogens in these patients.

While quantitative PCR was used as the method in this study, the primary study outcome was how patients were classified for chronicity of infection rather than the abundance of pathogen in samples. Because the study title does not refer to the comparison of diagnostic methods or the primary outcome, I would suggest a different title. Possibly: Q-PCR detection of cystic fibrosis pathogens increases the diagnosis of chronic lung infections.

Response: The Reviewer makes a fair point. The title was a source of considerable debate amongst the authors, with the original proposed title being not far from what the Reviewer has suggested. Therefore, we’ll change the title to “Bacterial culture underestimates lung pathogen detection and infection status in cystic fibrosis”.

2. The phrase “status disparity” in the running title sounds like a social comparison rather than a comparison of methods for classifying infections.

Response: Running title changed to now read “Infection detection disparity in CF”.

3. Subject Selection. There was some ambiguity in how the baseline infection status was defined, especially in comparing Discussion Line 157 with methods line 240.

Line 157 says patients were drawn from Pseudomonas-free clinics, containing patients with no new growth of P. aeruginosa for at least 12 months. Line 240 says that patients were eligible if considered free of

"chronic infection" as defined in reference 23. Could the authors provide baseline S. aureus and P. aeruginosa culture data for the 12 months preceding the study?

Response: To avoid confusion we have amended those two sentences to now read: At P7, L157: "In this study, patients were specifically drawn from *Pseudomonas*-free clinics, containing patients with no new growth of *P. aeruginosa* for at least 12 months and considered free of chronic infection by this pathogen." At P9, L240: "Patients were drawn from *Pseudomonas*-free clinics and were eligible if considered free of chronic infection with *Pseudomonas aeruginosa* and with no new growth of this pathogen for at least 12 months using diagnostic culture approaches and as defined by the Leeds criteria (23)." Hopefully, that now clears any confusion. We also thank the reviewer for their suggestion, but do not believe adding previous culture data will significantly add to the study.

4. Subject Selection. Figure 1 did not give the size of the center. It would be better to know how many subjects were excluded due to chronic P. aeruginosa or other exclusion criteria mentioned in the methods section such as age. The authors should also explain the rationale for excluding children under age 5.

Response: We have included the patient population size for each centre at P9, 238: "Adults were recruited from the Manchester Adult CF Centre (centre patient population $n = 466$ patients), and children from the Royal Manchester Children's Hospital ($n = 342$ patients). Patients of less than 5 were not suitable as typically they cannot produce sputum and are not suitable for LCI measurements.

5. Subject Characteristics. Table 1 did not report CFTR modulator therapy, which might affect quantitative recovery of pathogens or maybe even the incident detection of the pathogens. Azithromycin and Colistin were mentioned in the text. It was not given whether patients were receiving oral anti-Staphylococcal medications or other anti-Pseudomonal medications like inhaled tobramycin or aztreonam. Baseline infections for the two pathogens were not provided.

Response: In response to a comment above and to complement Table 1, we have created a new Table S3 that clearly details for the reader clinical characteristics of individual patients, including individual CFTR genotypes, lung function, etc.

6. Outcome definition. Because the main outcome of their assay is categorical - whether S. aureus or P. aeruginosa were detected, the authors need to provide information about how they defined a positive test. Was there a Ct value they considered positive? What was the limit of detection?

Response: We have already provided information on how culture based data was generated (from P10, L260) and how the QPCR based data was generated (moved from the Supplementary data and expanded upon in the Methods section)(from P10, L272). It is unclear which assay the reviewer referring to when they mention limit of detection? We assume from QPCR. We now have included from the QPCR based lower limit of detection in CFU ml-1 equivalents for each pathogen in the legends of Figure S1 and S2 as directed by Reviewer 1 below in Point10.

7. Quantitative results for the Q-PCR assay were presented in the supplement, given in CFU per mL equivalents. The methods section indicates the Q-PCR assay used pure strains as standards. I assume this is how CFU equivalents was determined. It is not clear how the sample volume was determined for cough swabs. Could the authors describe the calculation in greater detail?

Response: We provide detail on how we processed cough swabs differently from sputum at the DNA extraction stage to optimise recovery (from P10, L265). All samples were then processed in the same manner thereafter as detailed.

8. Were there quantitative differences in the Q-PCR assay when comparing culture-positive and culture-negative samples?

Response: That is an interesting question. But not something that was addressed in the analyses.

9. *The boxplots shown in the supplement were confusing. Perhaps the authors were trying to fit logarithmically-distributed data with whiskers that were defined from a linear-range calculation. The number of points above the upper whiskers in Figure S2 were far more than the number of points below the lower whiskers. In several plots in S1 and in the upper left-hand plot of S2 (P. aeruginosa in Pediatric Sputum), the upper whisker was below the 75th Percentile. Individual data points on this figure were on top of one another so that they couldn't be counted, especially in adults with sputum. Given the small number of patients, these points could be distributed better horizontally. I would suggest using GraphPad Prism or the beeswarm package in R so that individual data points are more apparent.*

10. *S1 figure was helpful because it gave the number of positive tests per subject. The total number of tests per subject can be found in table S2. The number of data points divided by the number of tests was equal to the height of the gray bars in Figure 2. The boxplots did not add to the figures for individual patients, especially when there were only 2 or 3 positive tests. It would be better if data points were outlined to make them easier to count. The limit of detection for the assay should be stated in the legend or on the figure. The S2 legend should explicitly state that it shows quantitation for positive tests only.*

11. *S2 figure - Pediatric P. aeruginosa panel. The legend gave mean {plus minus} an error estimate (I assume SD). The SD was larger than the mean, consistent with a skewed distribution. I don't think a statistical comparison is essential here, but the data distributions appeared different to me. The 75th percentile for cough swab was roughly the same as the 25th percentile for sputum. Since these data were not normally distributed, I suggest using Wilcoxon rank-sum test to compare these distributions rather than the Welch's t-test, if the authors want to make a statistical comparison.*

Response to 9, 10 & 11: We have redrawn Figures S1 and S2 to still include the scattergram element but have replaced the boxplots with line bars indicating the mean and standard deviation of the mean instead. In both figures we use the log₁₀ of the CFU ml⁻¹ equivalents and derive means and SDs from that. As directed, we have replaced Welch's test with the nonparametric Kruskal-Wallis test. The legends for these two supplementary figures have been amended (including addition of lower limit of QPCR detection for each pathogen). Also, related text within the manuscript has been updated. Based on the reviewer's queries, we believe this substantially improves the presentation of those supplemental figures. Thank you.

12. *Figure 2 displayed the percentage of positive samples as a stacked bar graph, stratified by patient, with children and adults divided on the x-axis. It was a nice visualization. I would recommend removing the minus signs on the y-axis for the molecular diagnostics.*

Response: Thank you for spotting that oversight. That has now been corrected.

13. *Figure 3 provides the same data as Figure 2 after applying > 0% and > 50% thresholds. It is not clear whether this classification scheme was based on test positivity over the entire duration of the study, or the final year of the study.*

Response: We have clearly indicated multiple times throughout the manuscript how chronic and intermittent infection status was defined using a modification of the Leeds criteria (Reference 23). For example in the Materials and methods and in the legend of Figure 1. We don't believe there is a need to repeat this again.

14. *The Figure 3 legend says "Changes in pathogen infection status from diagnostic culture to molecular-based detection." I think this could be rephrased, because the test didn't change whether the patient is actually infected - it changes how the authors would classify the infection.*

Response: This figure depicts the change in infection status when using diagnostic culture and then when defined by molecular-based detection. We are sorry this was not clear for the reviewer. We have amended

the first sentence in the Figure 3 legend to now read “Changes in pathogen infection status from when defined by diagnostic culture to then by molecular-based detection in paediatric and adult patients”.

15. *Results Lines 108. The authors make head-to-head comparisons of Q-PCR versus culture in terms of test positivity for the entire set of observations. The results are clearly different, but I would suggest using McNemar's test rather than Welch's t-test because the outcome they present is dichotomous (positive or negative) and the samples were paired.*

Response: Changed to Kruskal-Wallis based test along with other tests, as directed from above.

16. *Discussion Line 206 - There were a few instances where there was detection by culture but not by Q-PCR. Could the culture isolates be amplified by Q-PCR? Were there mutations affecting the ability to amplify the target genes?*

Response: Considering the data and the well-established primers that we used, the “mutations affecting the ability to amplify” was highly unlikely here. Looking at our data it would be more likely false positives from culture results, as explained in the text.

17. *The work is interesting and important, but there were some limitations that should be mentioned in the discussion:*

- a. While PCR testing allows detection of these pathogens, it has some limitations, like not allowing for antibiotic susceptibility testing.*
- b. PCR testing without sequencing does not allow the infecting strain to be identified and tracked. Culture methods would facilitate strain tracking.*
- c. The study did not enroll control subjects, making it hard to know how abnormal the CF measurements are versus the healthy population. It is possible that control individuals, under environmental exposure to *P. aeruginosa* would test positive by Q-PCR, yet they do not develop symptoms because their airways or immune systems effectively kill incident *P. aeruginosa*.*
- d. There is some potential for selection bias here, as the subjects studied by the authors were known not to be chronically infected. Thus, the patient selection enhances the apparent effect of Q-PCR detection. It would be good to know how many subjects in the center were not eligible for inclusion because they were already classified as chronically infected.*
- e. The supplement makes the claim that this study has "impact on clinical status for individual patients." I think that is an overstatement, as the authors provide no evidence that the clinical status (lung function, symptoms, exacerbations, etc) for individual patients has changed by virtue of this method of testing. It would be fair to say that changing the method of testing would impact how chronic infection status is classified.*

Response: There are multiple points raised here. However, many of these points pertain to pros and cons of culture and molecular approaches. From our response on this reviewer comment at the top of this review - On potential methodological factors, we acknowledge and discuss this in the Discussion from P8, L191 in the passage of text that starts “Although methodological strengths and weaknesses of molecular and culture detection methods have been reviewed by others (7, 10) it is important to consider potential factors that might contribute to the observed disparity between”. As pros and cons of methods have been extensively reviewed and discussed in detail elsewhere (including in reviews papers – references 7 and 10) we felt this was sufficient and kept to points we felt pertinent to this study.

We address the subpoints in turn here: WRT “*a. While PCR testing allows detection of these pathogens, it has some limitations, like not allowing for antibiotic susceptibility testing*”. That was never in the scope of this study, only pathogen detection between the two methods was.

WRT “*b. PCR testing without sequencing does not allow the infecting strain to be identified and tracked. Culture methods would facilitate strain tracking*”. Again, not in the scope of this study, only pathogen detection between two methods was.

WRT “c. *The study did not enroll control subjects, making it hard to know how abnormal the CF measurements are versus the healthy population. It is possible that control individuals, under environmental exposure to P. aeruginosa would test positive by Q-PCR, yet they do not develop symptoms because their airways or immune systems effectively kill incident P. aeruginosa.*” Again, From the outset, we have been clear what the study aims and what the study population was and why. Therefore, it is not clear what would be the need for a control group, how a control group would contribute/enhance the study, what would that control group be? Moreover, in the Discussion we do recognise (P9, L216) that for future studies.. “To further advance this work, future studies will need to have larger patient numbers representative of the wider CF population, encompassing, for example, differing disease severities, disease states, and exacerbation and infection histories”

WRT “d. *There is some potential for selection bias here, as the subjects studied by the authors were known not to be chronically infected. Thus, the patient selection enhances the apparent effect of Q-PCR detection. It would be good to know how many subjects in the center were not eligible for inclusion because they were already classified as chronically infected*”. This is the same point(s) as raised and addressed above at the top of this review.

WRT “e. *The supplement makes the claim that this study has "impact on clinical status for individual patients." I think that is an overstatement, as the authors provide no evidence that the clinical status (lung function, symptoms, exacerbations, etc) for individual patients has changed by virtue of this method of testing. It would be fair to say that changing the method of testing would impact how chronic infection status is classified.*” Again, this was raised and addressed above at the top of the review. In the discussion we consider and discuss possible clinical implications of infection based on not detecting infection that is present and misclassifying chronic infection status. Of course we do not know what the extent of clinical implications will be – hence talking specifically on possible clinical implications of infection. Again being cautious, we are careful to temper our language throughout, for example, by stating in the concluding section of the Discussion P9, from L229 “Again, future longitudinal studies with larger patient numbers will be needed to establish the extent of the clinical implications arising from this study. Moreover, such studies will be required to thoroughly assess targeted QPCR approaches before they could be approved for routine and widespread diagnostic use in clinical settings.” We have also amended a number of sentences, as detailed above, on this to further temper our language.

Reviewer #2 comments:

1. Clinical implications are unclear: The results of this study suggest that a large majority of people with CF are misclassified with underappreciation of chronic infection status. In children, based on culture, only one participant was classified as chronically infected with Pa whereas 10 (50%) were classified as chronic based on PCR (in adults, 0% by culture vs. 90% by PCR). While the authors note that qPCR has been found to be more sensitive than culture in prior studies (as they found), they note that clinical outcomes of these findings have not been studied. However, this current study doesn't add to the understanding of the clinical impact - was there a difference in exacerbation rate, lung function change for example in those who are identified as chronically infected with Pa by qPCR? For those who were intermittently infected with Pa - and presumably treated with inhaled antibiotics- was there a difference noted in the following culture (decrease in qPCR)?

Response: Focusing on clinical implications and clinical impact aspects of this comment - In response to comments by Reviewer#1 we acknowledge we do not know the extent of clinical implications from the study's observations. And as noted in response to Reviewer#1 we do discuss and consider possible infection based clinical implications in the Discussion. We are also careful not to overclaim and are cautious, being mindful to temper our language throughout, for example, by stating in the concluding section of the Discussion P9, from L229 “Again, future longitudinal studies with larger patient numbers will be needed to establish the extent of the clinical implications arising from this study. Moreover, such studies will be required to thoroughly assess targeted QPCR approaches before they could be approved for routine

and widespread diagnostic use in clinical settings". Informed by our findings, those future studies will help elucidate clinical impact. Moreover, from our response to Reviewer#1, we have tempered the language further by amending the following: Introduction P4, L81: we have replaced investigated with 'considered'. So sentence now reads: "Furthermore, none of those studies considered the impact that more sensitive detection methods would have on infection status or clinical outcomes". At P4, L90 at the end of the introduction the sentence now reads "Potential clinical implications for CF and for pathogen detection and surveillance more broadly are considered and discussed". Sentence at P7, L151 amended to include word 'potential. Now reads: "...with potential implications for understanding of infection in CF and clinical management of emergent infections". Sentence in Discussion at P9, L218 amended to include 'potential'. Now reads: "That said, it remains that our findings have potential implications for understanding the evolution of infection and the clinical care of patients".

2. Given the requirement that patients were negative for Pa at study entry, the patient group represents relatively healthy CF cohort, particularly for adults. Consistent with this, almost 50% of adults were pancreatic sufficient. What criteria were used for CF diagnosis? Did all patients have sweat chloride > 60? Were those with CFTR-related disorder included? Additional genotype information such as # with residual function mutations would be helpful in understanding the patient cohort. Were any on CFTR modulators during the study?

Response: In response to Reviewer#1 and to help the reader to better understand the cohorts, we have created a new Table in the Supplementary Materials (Table S3) that details the clinical characteristics of individual patients, including individual CFTR genotypes, lung function, etc.

3. While information about exacerbation status is provided in the supplement, was this considered in the analysis? What about acute antibiotic use?

Response: No, as this was not in the scope of the current study. This is something we would like to incorporate in the extended follow-up studies.

4. Figure S2 does not seem to match the results states. In the pediatric Pa panel, the abundance detected in OP appears lower than in sputum - the mean abundance in the legend and text says "7.93x10¹⁰ {plus minus} 1.86x10¹¹ (n = 24) and 5.38x10¹⁰ {plus minus} 2.14x10¹¹" for Pa in sputum versus OP. However visually, the mean of OP looks closer to 10⁷ vs 10⁹ for sputum. Similarly, the other panels don't seem to match the numbers given although OP / sputum look more similar in these groups. It's also interesting that the pediatric Pa quantities look higher than the adult Pa cultures - possibly due to more mild disease aggressiveness in adults.

Response: Thanks for highlighting this. We believe this has been subsequently resolved in response to Reviewer#1 where we have redrawn Figures S1 and S2 to make them clearer for the reader.

5. Figure S1 is a helpful graph. For Pa (less so for Sa), it appears that many patients/ samples had lower levels of Pa. For those with low Pa detected (e.g. < 10⁸) was qPCR detection more likely to be concordant with culture?

Response: Not typically. Figure 1 also highlights those disparities between culture and molecular-based approaches.

Minor comments:

1. In the supplement, table S2, dates are provided for samples collected from individual patients. Suggest changing this to baseline, follow-up time in months rather than exact dates to avoid the risk of identifying individuals.

Response: Thank you for the suggestion. We have discussed this with the clinicians on this study. Subsequently, it was judged there would be negligible to no risk. Hence, we will keep as is.

2. Table S1 is an outstanding summary of the current literature and the attending to detail is appreciated.

Response: We thank the reviewer for their kind words and recognition. As stated to Reviewer#1, we felt it important to include this to contextualise our study with previous work.

3. The use of %'s in addition to numbers throughout the results section would help the reader.

Response: Thank you for the suggestion. However, we have already done just that were appropriate in the Results section.

4. Figure S2: Why are the individual dots separated along the x-axis within sample groups?

Response: This has been now remedied as a result of redrawing both Figures S1 and S2 and using log₁₀ CFU ml⁻¹ equivalents.

June 22, 2022

Prof. Christopher van der Gast
Manchester Metropolitan University
Department of Life Sciences
John Dalton Building
Chester Street
Manchester M1 5GD
United Kingdom

Re: Spectrum00419-22R1 (Bacterial culture underestimates lung pathogen detection and infection status in cystic fibrosis)

Dear Prof. Christopher van der Gast:

Thank you for submitting your revised manuscript to Microbiology Spectrum. The paper was re-reviewed by the original reviewers and myself. You will see there was not a clear agreement between these the two reviewers (below). And I also am still having a few problems with the paper. I believe there is a lot of interesting data included in this manuscript (I still really like Table S1), but much of it is lost because of the manner in which it is presented and also some of the terminology that is used. These are the main points that I think still need to be addressed:

(1) While these patients were "Pseudomonas-free" at the outset (as determined by culture), *Pseudomonas aeruginosa* was detected by QPCR. Would your data allow you to determine the CFU ml⁻¹ equivalents when colonies are first detectable by the clinical micro lab? This would seem to be very important information for the field.

Can you use your data to determine how many CFU ml⁻¹ equivalents of *P. aeruginosa* and/or *S. aureus* are present during an exacerbation by combining the data you have with the information from Table S2? This would also seem to be something very important know and exciting data to present.

(2) The Leeds criteria defines chronic infection by the "gold standard" of culture. It seems that just detecting "using QPCR" and then defining this as a "chronic infection" does a disservice to the community, as it implies much more about the clinical condition of the patients (as well as the adaptation/phenotype of the isolates) than is actually known. Perhaps another term should be used throughout this paper. Maybe "no detection vs. intermittent detection vs. consistent (or continuous) detection" would be better since it doesn't assume anything other than the detection of the QPCR product. I think your conclusions, for the most part, will remain the same, but the terminology would not allow your results to be over interpreted or taken out of context (in other words, suggesting inappropriate treatment for a condition that is not necessarily "chronic"). With this in mind, I do not think Figure 2 is necessary or adds much to the story.

(3) Finally, I am having a hard time with some of the numbers and the verbiage in the Abstract (and where it is repeated in the text): "1 to 28 and 9 to 34, respectively". What does that refer to? If this is *Pseudomonas* and *Staphylococcus*, why does it say "or" in the phrase beforehand. The next sentence is similarly confusing. Did the "infection status change" or did the "classification of the infection status change"?

I actually believe these changes will help address Reviewer #2's issues and also serve to make this a much stronger paper.

Joanna

Link Not Available

ASM policy requires that data be available to the public upon online posting of the article, so please verify all links to sequence records, if present, and make sure that each number retrieves the full record of the data. If a new accession number is not linked or a link is broken, provide production staff with the correct URL for the record. If the accession numbers for new data are not publicly accessible before the expected online posting of the article, publication of your article may be delayed; please contact

the ASM production staff immediately with the expected release date.

Sincerely,

Joanna Goldberg

Journals Department
Reviewer comments:

Reviewer #1 (Comments for the Author):

The authors' data supports the conclusions. All previous comments were addressed. Thank you for sharing this important work.

Reviewer #2 (Comments for the Author):

I appreciate the authors response to the reviewers comments, but I remain concerned about the specificity of the qPCR test. The investigators report that while 1/40 patients were chronically infected by *Pseudomonas aeruginosa* based on culture, using qPCR 28/40 (70%) would be classified as chronically infected. The qPCR method used- detection of the *oprL* gene, has been shown to have a high sensitivity but specificity is lower (70% is cited in this paper). In this population with relatively mild disease and a low incidence of *Pa* infection (based on culture which is our current gold standard), the positive predictive value will be low. The investigators do not include a discussion of limitations of qPCR detection or poor specificity in the discussion and the conclusions suggest that we are misclassifying a large number of patients with implications for treatment.

Staff Comments:

Preparing Revision Guidelines

Please return the manuscript within 60 days; if you cannot complete the modification within this time period, please contact me. If you do not wish to modify the manuscript and prefer to submit it to another journal, please notify me of your decision immediately so that the manuscript may be formally withdrawn from consideration by Microbiology Spectrum.

If your manuscript is accepted for publication, you will be contacted separately about payment when the proofs are issued;

please follow the instructions in that e-mail. Arrangements for payment must be made before your article is published. For a complete list of **Publication Fees**, including supplemental material costs, please visit our website.

Response to Reviewers

Re: Spectrum00419-22R1 (Bacterial culture underestimates lung pathogen detection and infection status in cystic fibrosis)

Dear Professor Goldberg,

We thank you and the reviewers for the time taken so far in the review of our manuscript. Previously, we thoroughly addressed all comments from the reviewers and from you, the Editor. Here, we address the comments and suggestions from yourself and the reviewers in turn below. For clarity, reviewer comments are in italics with our response directly below a given comment. Starting with “**Response:** ...” in each instance. Changes made in the Marked-Up Manuscript are in red coloured text. Page and Line numbers used below refer to those in the Marked-Up Manuscript.

Editor comments:

Thank you for submitting your revised manuscript to Microbiology Spectrum. The paper was re-reviewed by the original reviewers and myself. You will see there was not a clear agreement between these the two reviewers (below). And I also am still having a few problems with the paper. I believe there is a lot of interesting data included in this manuscript (I still really like Table S1), but much of it is lost because of the manner in which it is presented and also some of the terminology that is used. These are the main points that I think still need to be addressed:

Response: We are somewhat surprised and confused to find brand new comments from the Editor and Reviewer#2 that had not previously been raised in the original review. Essentially, this makes for new reviews with new comments to address. The way these new comments are phrased indicates they are ‘concerns’ the Editor and Reviewer#2 had all along and had not arisen from the subsequent wide-ranging changes/edits made resulting for the original reviews. While these new comments/concerns will not change the overarching findings of the study, we believe this is not particularly fair or helpful. It does not help the early career researchers on this paper who are dependent on getting this and other linked papers published in a timely manner. We believe there is nothing controversial about our study. It fits the remit of *Microbiology Spectrum*, will be of interest to the CF microbiology community, and we have done our utmost to accommodate the review comments from both the reviewers and editor, which have helped improve this manuscript. Further and with due respect, in our combined experience an Editor actively reviewing a manuscript in this manner is unusual. That said, we provide robust, yet respectful, rebuttal to the Editor’s new review comments below and then go onto do the same for Reviewer#2.

(1) While these patients were "Pseudomonas-free" at the outset (as determined by culture), Pseudomonas aeruginosa was detected by QPCR. Would your data allow you to determine the CFU ml⁻¹ equivalents when colonies are first detectable by the clinical micro lab? This would seem to be very important information for the field.

Can you use your data to determine how many CFU ml⁻¹ equivalents of P. aeruginosa and/or S. aureus are present during an exacerbation by combining the data you have with the information from Table S2? This would also seem to be something very important know and exciting data to present.

Response: We address these two points in turn: For the first comment (1a) - We have considered this, and measures of QPCR abundance in this context do not show anything of note for either pathogen that would change or enhance the findings / outcomes of this study. For *S. aureus*: Minimum = 1.27×10^4 CFU ml⁻¹ equivalents; Maximum 2.36×10^{11} CFU ml⁻¹ equivalents; Mean \pm SD = $4.84 \times 10^9 \pm 5.34 \times 10^2$ CFU ml⁻¹ equivalents. For *P. aeruginosa*: Minimum = 6.86×10^5 CFU ml⁻¹ equivalents; Maximum = 1.27×10^{11} CFU ml⁻¹ equivalents; Mean \pm SD = $9.86 \times 10^9 \pm 1.9 \times 10^2$ CFU ml⁻¹ equivalents.

On the second comment (1b) – That is a great suggestion. In fact that is the follow-up study which we have completed and is almost ready to be submitted for peer-review. In brief, we follow adult patients through antibiotic treatment for pulmonary exacerbation (at start of treatment, end of treatment, and baseline follow-up sampling). We use QPCR to target multiple canonical CF pathogens (including, *P. aeruginosa*, *S. aureus*, *Burkholderia cepacia* complex members, *Stenotrophomonas maltophilia*, and *Achromobacter xylosoxidans*). Further supported by 16S rRNA gene amplicon sequencing of the whole bacterial microbiota in each sample. The purpose of the study was to assess pathogen abundance as an outcome measure for pulmonary exacerbation treatment outcomes.

(2) The Leeds criteria defines chronic infection by the "gold standard" of culture. It seems that just detecting "using QPCR" and then defining this as a "chronic infection" does a disserve to the community, as it implies much more about the clinical condition of the patients (as well as the adaptation/phenotype of the isolates) than is actually known. Perhaps another term should be used throughout this paper. Maybe "no detection vs. intermittent detection vs. consistent (or continuous) detection" would be better since it doesn't assume anything other than the detection of the QPCR product. I think your conclusions, for the most part, will remain the same, but the terminology would not allow your results to be over interpreted or taken out of context (in other words, suggesting inappropriate treatment for a condition that is not necessarily "chronic"). With this in mind, I do not think Figure 2 is necessary or adds much to the story.

Response: Using alternative terms to those used in the Leeds criteria will not be helpful and will only serve to confuse readers. In the manuscript it is clear we are comparing pathogen detection using both culture and targeted QPCR. In order to compare between those detection approaches it would be confusing to use different terminology for infection status classification, i.e. using 'chronic infection' for culture-based data but 'consistent detection' by QPCR. Hence parity in terminology is important. However, to help clarify further, and in response to the next comment from the Editor (Comment 3), we change text as appropriate to make clear/emphasise we refer to classification of infection status. Moreover, we take it that Figure 3 is being referred to and not Figure 2. That figure (Figure 3) is important in the context of visualising changes in infection status classification and is crucial to the study narrative.

Finally, we respectfully refute the assertion that we seek to over interpret our results or do a "disserve to the community". If anything, we dispassionately present our findings and carefully temper our language (in the original draft and then again following peer review) along with clearly pointing out in the discussion / conclusion the need for future work. Moreover, our three experienced CF respiratory consultant clinician co-authors all agree our work could not be "taken out of context" and that do not at any point suggest "inappropriate treatment for a condition".

(3) Finally, I am having a hard time with some of the numbers and the verbiage in the Abstract (and where it is repeated in the text): "1 to 28 and 9 to 34, respectively". What does that refer to? If this is Pseudomonas and Staphylococcus, why does it say "or" in the phase beforehand. The next sentence is similarly confusing. Did the "infection status change" or did the "classification of the infection status change"?

Response: For the benefit of the reader (and Editor) we now emphasise "classification of the infection status" when assess by culture or by QPCR. We have made the following changes (red text) to existing text:

Abstract At Page 2, from Line 34 now reads: "Classification of infection status also significantly altered in both paediatric and adult patients, where the number of patients deemed chronically infected with *Pseudomonas* and *Staphylococcus* increased from 1 to 28 and 9 to 34, respectively. Overall, *Pseudomonas* and *Staphylococcus* infection status classification changed respectively for 36 and 27 of all patients. In no cases did molecular identification lead to a patient being in a less clinically serious infection category. Pathogen detection and infection status classification significantly increased when assessed by QPCR in comparison to culture."

At Page 5 from Line 114 we have amended the text to read : “More specifically, *P. aeruginosa* and *S. aureus* were detected by culture on at least one occasion in 12 (30%) and 29 (73%) of the patients, respectively. Conversely, both pathogens were detected in all 40 patients on at least one occasion by QPCR (Figure 2). For *P. aeruginosa*, culture determined only one patient to be **classified as** chronically infected and a further 11 **classified as** intermittently infected (Figures 2 & 3). That increased to 28 **classified as** chronically infected and 12 **classified as** intermittently infected with *P. aeruginosa* when using QPCR to identify presence of infection. For *S. aureus*, nine patients were deemed to be chronically infected by culture compared to 34 **patients classified as chronically infected by** QPCR. Change in bacterial infection status **classification** was seen in both the adult and paediatric patients. Overall, *P. aeruginosa* and *S. aureus* infection status **classification** changed for 36 and 27 out of all patients, respectively. In no cases did molecular identification lead to a patient being allocated to a negative or intermittent infection category when culture placed them **with an** intermittent or chronically infected **classification** (Figure 3).”

Figure 3 Legend has been amended to read: “**Figure 3** Changes in pathogen infection status **classification** from when defined by diagnostic culture to then by molecular-based detection in paediatric and adult patients. Given are changes in infection status **classification** for (a) *Pseudomonas aeruginosa* and (b) *Staphylococcus aureus* in each of the children and adult CF patients. In each instance, coloured lines represent individual patients.”

Reviewer comments:

Reviewer #1:

The authors' data supports the conclusions. All previous comments were addressed. Thank you for sharing this important work.

Response: We are grateful to the Reviewer for their time and consideration. We are also very grateful they recognise this as important work.

Reviewer #2:

*I appreciate the authors response to the reviewers comments, but I remain concerned about the specificity of the qPCR test. The investigators report that while 1/40 patients were chronically infected by *Pseudomonas aeruginosa* based on culture, using qPCR 28/40 (70%) would be classified as chronically infected. The qPCR method used- detection of the oprL gene, has been shown to have a high sensitivity but specificity is lower (70% is cited in this paper). In this population with relatively mild disease and a low incidence of Pa infection (based on culture which is our current gold standard), the positive predictive value will be low. The investigators do not include a discussion of limitations of qPCR detection or poor specificity in the discussion and the conclusions suggest that we are misclassifying a large number of patients with implications for treatment.*

Response: Likewise, we are grateful to the Reviewer for their time and consideration. Reviewer#2 did not raise any of this on their previous review. However, Reviewer#1 did make passing mention to including “some of the inherent limitations of molecular diagnostics”. We subsequently responded to Reviewer#1 that “On potential methodological factors, we do acknowledge and discuss this in the Discussion from P8, L191 in the passage of text that starts “Although methodological strengths and weaknesses of molecular and culture detection methods have been reviewed by others (7, 10) it is important to consider potential factors that might contribute to the observed disparity between As pros and cons of methods have been extensively reviewed and discussed in detail elsewhere (including in reviews papers – references 7 and 10) we felt this was sufficient and kept to points we felt pertinent to this study”.

Also, both Reviewer#1 and Reviewer#2 did not previously raise any concerns about PCR primer specificity.

As noted above in response to the Editor, this is a new review comment that has not been previously raised to us. Although it does not change the outcome of the study, we consider at this stage this to be somewhat unfair and unhelpful.

Regardless, it is not immediately clear what point they are trying to make about specificity and what we are meant to do. We are not sure if they are trying to state that the qPCR approach used for *P. aeruginosa* is highly sensitive and that is why it is detecting more pathogen presence across samples compared to diagnostic culture? If so, then yes this would be an *a priori* assumption as it has been widely found and is widely accepted that qPCR is more sensitive than culture in prior studies (and acknowledged in this manuscript and as highlighted in Table S1 Literature Review). Moreover on sensitivity, in response to a Reviewer#1 comment (Point10 in the original review) we included the lower limits of detection for each pathogen in the legends for Figures S1 and S2 – “Lower detection limits derived from standard curves for *P. aeruginosa* and *S. aureus* were 2.2×10^2 and 5.2×10^3 CFU ml⁻¹ equivalents, respectively”. So, if *P. aeruginosa* or *S. aureus* was present in a sample above their respective detection threshold it would be expected that they would be detected by QPCR. The disparity between culture- and non-culture-based detection methods, as we posit and discuss in detail in the discussion, was most likely related to factors affecting detection by culture. For example, bacteria going into a viable but not culturable (VBNC) state, as indicated here by the target pathogens not being detected by culture but detected by molecular means, is a well-known and long accepted phenomenon. This has certainly been long established in environmental microbiology but perhaps not in clinical microbiology? Hopefully, combined this provides clarity here.

July 31, 2022

Prof. Christopher van der Gast
Manchester Metropolitan University
Department of Life Sciences
John Dalton Building
Chester Street
Manchester M1 5GD
United Kingdom

Re: Spectrum00419-22R2 (Bacterial culture underestimates lung pathogen detection and infection status in cystic fibrosis)

Dear Prof. Christopher van der Gast:

Your manuscript has been accepted, and I am forwarding it to the ASM Journals Department for publication. You will be notified when your proofs are ready to be viewed.

Sincerely,

Joanna Goldberg
Editor, Microbiology Spectrum
